# Female reproductive history in relation to chronic obstructive pulmonary disease and lung function in UK biobank: a prospective population-based cohort study

Rosalind Tang [1,2] Abigail Fraser,[3,4] Maria Christine Magnus[3,4,5]

For numbered affiliations see end of article.

**Correspondence to**
Rosalind Tang;
rt16358@bristol.ac.uk

## ABSTRACT

**Objectives** Sex differences in respiratory physiology and predilection for developing chronic obstructive pulmonary disease (COPD) have been documented, suggesting that female sex hormones may influence pathogenesis. We investigated whether aspects of female reproductive health might play a role in risk of COPD among women.

**Design** Population-based prospective cohort study.

**Setting** UK Biobank recruited across 22 centres in the UK between 2006 to 2010.

**Primary and secondary outcomes measures** We examined a range of female reproductive health indicators in relation to risk of COPD-related hospitalisation/death (n=271 271) using Cox proportional hazards regression; and lung function (n=273 441) using linear regression.

**Results** Parity >3 was associated with greater risk of COPD-related hospitalisation/death (adjusted HR 1.45; 95% CI: 1.16 to 1.82) and lower forced expiratory volume at 1 second/forced vital capacity ratio ($FEV_1$/FVC) (adjusted mean difference −0.06; 95% CI: -0.07 to 0.04). Any oral contraception use was associated with lower risk of COPD-related hospitalisation/death (adjusted HR 0.85; 95% CI: 0.74 to 0.97) and greater $FEV_1$/FVC (adjusted mean difference 0.01; 95% CI: 0.003 to 0.03). Late menarche (age >15) and early menopause (age <47) were also associated with greater risk of COPD-related hospitalisation/death (but not lung function), while endometriosis was associated with greater $FEV_1$/FVC (not COPD-related hospitalisation/death). Early menarche (age <12 years) was associated with lower $FEV_1$/FVC (but not COPD hospitalisation/death). Associations with polycystic ovary syndrome (PCOS) or ovarian cysts, any hormone replacement therapy (HRT) use, hysterectomy-alone and both hysterectomy and bilateral oophorectomy were in opposing directions for COPD-related hospitalisation/death (greater risk) and $FEV_1$/FVC (positive association).

**Conclusions** Multiple female reproductive health indicators across the life course are associated with COPD-related hospitalisation/death and lung function. Further studies are necessary to understand the opposing associations of PCOS/ovarian cysts, HRT and hysterectomy with COPD and objective measures of airway obstruction.

## Strengths and limitations of this study

► This large, population-based, prospective cohort study is the first to assess chronic obstructive pulmonary disease (COPD) risk with age at menarche, menopause status, hysterectomy, gynaecological surgery, endometriosis or polycystic ovary syndrome.

► No study has examined a wide range of reproductive health indicators, spanning the reproductive and post-reproductive periods, with regards to COPD.

► We included a comprehensive list of potential confounders, such as age, height, body mass index, ethnicity, education, household income, Townsend deprivation index, smoking history, asthma, maternal COPD and paternal COPD.

► We relied on participants to self-report reproductive history and we cannot exclude a potential influence of selection bias, due to the low participation rate in UK Biobank.

► Data on environmental tobacco smoke were not available and we cannot exclude residual bias from this exposure.

## INTRODUCTION

There are clear sex differences in lung anatomy and physiology throughout the life course.[1] This is also reflected in the risk of obstructive lung disease. For example, boys have greater risk of asthma during early childhood compared with girls, but this sex difference reverses after puberty, when girls are found to have the greater risk of new-onset asthma.[2] Chronic obstructive pulmonary disease (COPD) has traditionally been seen as a disease predominantly affecting men. However, in more recent years, the overall burden of the disease has shifted more towards women.[2 3] Women with COPD also have, on average, fewer pack-years of smoking history than men with COPD; and non-smokers with COPD are more often

female.[1] This points towards a greater underlying susceptibility among women, and highlights the importance of understanding female-specific risk factors for COPD.

Female sex hormones are putative mediators of respiratory health, and regulate bronchodilation, cell proliferation, inflammation and metabolism of toxic cigarette smoke-related metabolites.[1 4–7] Notably, oestrogen and progesterone appear to have varying effects on lung function across the lifespan.[1] We know little, however, about female reproductive health indicators in relation to COPD risk.[1] These reproductive health indicators include markers of endogenous hormone exposure (age at menarche, age at menopause and length of reproductive lifespan), exogenous hormone exposure (oral contraceptive use and hormone replacement therapy), as well as pregnancy history. Of two existing studies, one study found no evidence of an association between hormone replacement therapy (HRT) and COPD risk,[8] while another reported a positive association between early age at first pregnancy (<20 years) and COPD-related mortality.[9] There is more evidence regarding the relationship between female reproductive health indicators and lung function. Early menarche,[10–12] nulliparity,[13] younger age at first pregnancy[12] and postmenopausal status[14–19] have each been associated with reduced lung function. Evidence of the effects of HRT is contradictory.[14 15 20 21] In the present study, we therefore investigate associations of multiple female reproductive health indicators across the lifespan with incidence of COPD hospitalisation/death and lung function in the UK Biobank cohort.

## METHODS
### Study population
Our study included female participants in UK Biobank (figure 1), a population-based cohort study which recruited men and women age 40 to 69 years across 22 centres in England, Wales and Scotland between 2006 to 2010.[22] The response rate was 5.5%.[22] All participants provided written informed consent. Data on 273 441 women were available after withdrawals. At baseline, participants self-reported sociodemographic information, family medical history, early-life exposures, psychosocial history and medical history. Physical measurements included spirometry. Information on health outcomes after recruitment into the cohort was gathered prospectively via data linkages with hospital and death registers. Register data were available up to 28 February 2015 for England, 16 March 2015 for Wales and 28 October 2014 for Scotland. UK Biobank does not require separate ethics approvals for specific analyses. A generic Research Tissue Bank approval for access to UK Biobank data and/or samples has been recommended by the National Research Ethics Service and has been obtained from the governing UK Biobank Research Ethics Committee (http://www.ukbiobank.ac.uk/wp-content/uploads/2012/09/Access-Procedures-2011-1.pdf). Applications

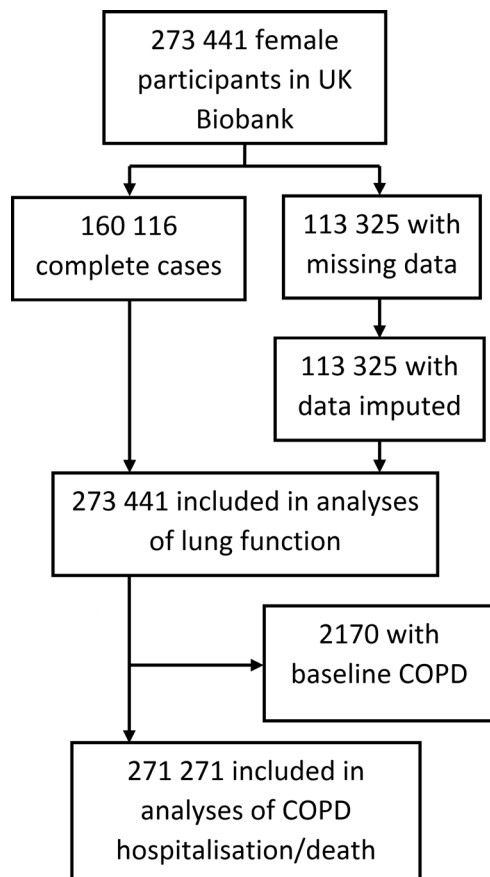

**Figure 1** Flow chart of study population. COPD denotes chronic obstructive pulmonary disease. Missing data is in one or more of the following variables: height, ethnicity, educational attainment, income, Townsend deprivation index, smoking history, spirometry measurements, maternal COPD history, paternal COPD history, age at menarche, menopause status, age at natural menopause, parity, oral contraception use, hormone replacement therapy use or gynaecological surgery history.

to access the resource are approved by the UK Biobank Access Sub-Committee and Ethics & Governance Council.

### Female reproductive health indicators
We investigated the following self-reported female reproductive health indicators: age at menarche (<12, 12 to 15 and >15 years), menopause status (no, yes), age at natural menopause (<47, 47 to 49, 50 to 52 and >52 years), parity (0, 1, 2, 3 and >3), history of polycystic ovary syndrome (PCOS) or ovarian cysts, history of endometriosis, oral contraception (OC) use (never/ever), years of OC use (0, 1, 2 to 4, 5 to 9, 10 to 15 and >15), hormone replacement therapy (HRT) use (never/ever), years of HRT use (0, 1 to 2, 3 to 5, 6 to 10 and >10) and gynaecological surgery (hysterectomy, oophorectomy and hysterectomy with bilateral oophorectomy). We calculated years of OC and HRT use as the differences between age at first and last use, and created categories according to quintiles.

Amaral *et al* previously reported decreased lung function in postmenopausal women in UK Biobank.[16] We

repeat these analyses for completeness alongside analyses of other female reproductive health indicators.

### Respiratory outcomes

Prevalent COPD at baseline was identified using self-report and registrations before baseline in the hospital registers. Incident hospitalisations/deaths from COPD after baseline were identified using hospital and death registers. The following International Classification of Diseases (ICD) codes were used to capture COPD: ICD-9 codes 490 to 492, 494 and 496, or ICD-10 codes J40 to J44.

Forced expiratory volume at 1 second ($FEV_1$) and forced vital capacity (FVC) were measured using Vitalograph Pneumotrac 6800 (Vitalograph, UK) at baseline. Spirometry was not performed if participants had experienced a chest infection in the last month; had a lifetime history of detached retina or collapsed lung; had experienced a heart attack, eye surgery or surgery to the chest or abdomen in the last 3 months; were currently pregnant in the first or third trimester or were currently using medication for tuberculosis. Two measurements were conducted if the difference between these measurements was ≤5%. A third measurement was conducted if the difference was >5%. We used the single greatest measurements of $FEV_1$ and FVC per participant. Post-bronchodilator spirometry was not available, although drug treatment was not withheld. Spirometry measurements were converted to internally standardised z-scores by age and height based on the Global Lung Initiative 2012 recommendations[23] to reduce bias related to age. These z-scores were used for all analyses.

### Statistical analyses

We used Cox proportional hazards regression to examine the association of each female reproductive health indicator in relation to incidence of COPD hospitalisation/ death. Participants without COPD at baseline were followed from enrolment until hospitalisation/death from COPD, death from other causes or the end of follow-up if the participants were still alive and disease-free. The time axis for the Cox regression was calendar time. We examined deviations from the proportional hazard assumption using Schoenfeld residuals.[24] Age-adjusted and confounder-adjusted estimates are reported as HR with 95% CI. The following potential confounders were included in multivariable analyses: age, height, log-transformed body mass index (BMI) at baseline, ethnicity (White or non-White), educational attainment (college, university or other professional degree; Advanced levels, Advanced Subsidiary levels or equivalent; Ordinary levels, General Certificate of Secondary Educations or equivalent; Certificate of Secondary Educations or equivalent; National Vocational Qualifications, Higher National Diploma, Higher National Certificate or equivalent; other), annual household income (<£18 000, £18 001 to £31 000, £31 001 to £52 000, £52 001 to £100 000, >£100 000), Townsend deprivation index, smoking history in pack-years (none, <11, 11 to 20, 21 to 30, >30),

and parental history of COPD. We performed sensitivity analyses of (1) menopause status and age at natural menopause with further adjustment for HRT; (2) menopause status excluding participants age <45 years or >60 years at baseline to reduce potential bias by age; (3) all reproductive health indicators after excluding participants with respiratory conditions (ie, COPD, asthma, tuberculosis or pneumonia) reported at baseline or $FEV_1$/FVC<0.7 at baseline, as they possibly had undiagnosed COPD[25]; (4) all reproductive health indicators with smoking history adjusted as two covariates (duration in years and average number of cigarettes/day) rather than one covariate (pack-years); (5) all reproductive health indicators with further adjustment for baseline smoking status (never, previous, current); (6) all reproductive health indicators after excluding participants with any history of smoking and (7) all reproductive health indicators with further adjustment for other comorbidities (cardiovascular disease or diabetes) reported at baseline.

We used linear regression in the analysis of standardised spirometry measurements. The multivariable linear regression models were adjusted for the same covariates as for hospitalisation/death from COPD, with the addition of asthma at baseline. We reported results as mean difference in the spirometry measurements z-scores with 95% CI. These mean differences in z-scores can be converted to raw values by multiplication with the SD in spirometry measurement for the given age and height group. Sensitivity analyses for standardised spirometry measurements were also conducted with: (1) menopause status excluding participants age <45 years or >60 years at baseline, (2) smoking history adjusted as two covariates (duration in years and average number of cigarettes/ day), (3) further adjustment for baseline smoking status and (4) further adjustment for comorbidities baseline (cardiovascular disease or diabetes).

We performed multiple imputation by chained equations to account for missing data. Exposures, covariates and outcomes were included in the imputation model. We imputed 20 data sets, which were then analysed using Rubin's rules.[26] Details of the imputation model are given in the online supplementary file table S1. All analyses were performed using Stata/MP 15 (StataCorp LLC, College Station, USA). Sensitivity analyses of the complete cases was performed for comparison with our imputed data.

### Patient and public involvement

The present study did not involve patients and the public.

### RESULTS

Of the 273 441 women included in this study, 2170 had prevalent COPD at baseline (figure 1). These prevalent cases were excluded from the analysis of incident COPD-related hospitalisation/death. By the end of follow-up (median 6 years), there were 1138 incident cases of hospitalisation/death from COPD, yielding an incidence rate of 0.19 per 100 000 person years.

In UK Biobank, 28.4% of women did not have spirometry data, 16.1% were uncertain of or did not report menopause status and 14.3% did not report smoking history. The proportion of missing information on the other covariates were modest (table 1). Overall, women with spirometry data were younger, more often White, had higher educational attainment, higher annual household income, less deprivation (lower Townsend index), greater smoking history and were more likely to have asthma than women who did not undergo spirometry assessment (online supplementary file table S2). The distribution of participants' characteristics across the imputed data sets is shown in table 1.

Late (age >15 years) menarche was associated with increased risk of hospitalisation/death from COPD compared with menarche between age 12 to 15. The multiple-adjusted HR was 1.37 (95% CI: 1.11 to 1.71) (table 2). There was a modest association between early menarche and COPD hospitalisation/death (HR 1.15; 95% CI: 1.00 to 1.32). Early menarche (age <12 years) was associated with lower $FEV_1/FVC$ (table 3).

Risk of hospitalisation/death from COPD did not differ between premenopausal and postmenopausal women (table 2), and this finding was similar after further adjusting for HRT use (HR 1.02; 95% CI: 0.78 to 1.35). In examining age at menopause and risk hospitalisation/death from COPD among women who experienced natural menopause (n=159 571), the risk was higher in women who experienced menopause age <47 years (HR 1.44; 95% CI: 1.19 to 1.75) when compared with age 50 to 52 (table 2). This association remained after further adjustment for HRT use (HR 1.43; 95% CI: 1.18 to 1.74). Postmenopausal status at baseline and early natural menopause were each associated with lower $FEV_1$ and FVC but not $FEV_1/FVC$ (table 3). These associations did not change after further adjustment for HRT ever use (results not shown). After restricting the sample to participants who were age 45 to 60 years (inclusive) at the time of reporting menopause status to reduce residual bias due to age, we found that menopause status was positively associated with COPD-related hospitalisation/death (HR 1.57; 95% CI: 1.07 to 2.30), while associations with spirometry measures did not change (results not shown).

Hospitalisation/death from COPD did not differ between parous and nulliparous women, but parity >3 was associated with increased risk (HR 1.45; 95% CI: 1.16 to 1.82) compared with nulliparity. Women who were nulliparous had lower $FEV_1$ and FVC but greater $FEV_1/FVC$ compared with parous women, while women with parity >3 had greater $FEV_1$ and FVC but lower $FEV_1/FVC$ compared with nulliparous women (table 3).

A history of PCOS or ovarian cysts was associated with increased risk of hospitalisation/death from COPD (HR 1.61; 95% CI: 1.12 to 2.32) and greater $FEV_1/FVC$ (mean difference 0.05; 95% CI: 0.02 to 0.09), while there was no evidence of an association with $FEV_1$ or FVC. A history of endometriosis was not associated with COPD hospitalisation/death, $FEV_1$ or FVC (tables 2 and 3), though we observed a modest positive association with $FEV_1/FVC$ (mean difference 0.07; 95% CI: 0.03 to 0.10).

We observed several associations between exogenous hormone use and COPD hospitalisation/death. Women who had ever used OC had lower risk of hospitalisation/death from COPD compared with never users (HR 0.85; 95% CI: 0.74 to 0.97) (table 2). This appeared to be driven by OC use of >9 years (table 2). In line with the observed findings for COPD, OC ever use and duration of use showed positive associations with $FEV_1$, FVC and $FEV_1/FVC$ (table 3).

Ever using HRT was associated with a modest increased risk of hospitalisation/death from COPD (HR 1.15; 95% CI: 1.01 to 1.30) (table 2). We observed no associations of HRT ever use with $FEV_1$ or FVC, but a positive association with $FEV_1/FVC$ (table 3). When we examined the duration of HRT use, we only observed increased hospitalisation/death from COPD (HR 1.33; 95% CI: 1.12 to 1.57) and greater $FEV_1/FVC$ (mean difference 0.02; 95% CI: 0.01 to 0.04) with 1 to 2 years of use (tables 2 and 3). There was no evidence of a dose-response relationship.

A history of hysterectomy (with or without bilateral oophorectomy), but not bilateral oophorectomy-alone, were associated with greater risk of COPD hospitalisation/death (table 2). At the same time, hysterectomy was associated with higher, rather than lower, $FEV_1/FVC$ (table 3). Hysterectomy with bilateral oophorectomy was also associated with lower FVC (table 3).

The directions of association between individual female reproductive health indicators with COPD risk and spirometry measures are summarised in table 4. Either a respiratory illness (asthma, COPD, tuberculosis and/or pneumonia) and/or $FEV_1/FVC<0.7$ at baseline was reported by 55 929 women. However, our findings after excluding these women from our analyses of COPD-related hospitalisation/death were similar to our main findings (online supplementary file table S3).

We report the results of our sensitivity analyses, after (1) further adjusting for baseline smoking status; (2) adjusting for smoking history as two covariates including duration in years of smoking and average number of cigarettes per day; (3) excluding women who have ever smoked or (4) further adjusting for cardiovascular disease or diabetes reported at baseline, in the online supplementary file tables S4-S8. The direction and magnitude of associations remained similar for risk of COPD hospitalisation/death (online supplementary file tables S4 and S6) and change in spirometry measures (online supplementary file tables S7 and S8) after further adjustment for the above covariates. In our analyses of never smokers (n=161 626), however, associations of COPD-related hospitalisation/death with early menopause, parity >3, PCOS/ovarian cysts, ever using oral contraception or hormone replacement therapy became null (online supplementary file table S5). Early age at menarche (<12 years) was also positively associated with COPD-related hospitalisation/death in this restricted sample of never smokers (HR 1.39; 95% CI: 1.02 to 1.90), while the

**Table 1** Distribution of observed and imputed data in UK biobank women at baseline (n=273 441)

| Characteristic | % data imputed | Mean (SE) or % in observed data | Mean (SE) or % in imputed data |
|---|---|---|---|
| Age (years) | 0 | 56.35 (0.02) | 56.35 (0.02) |
| <55 | | 39.5 | 39.5 |
| 55–59 | | 18.6 | 18.6 |
| 60–64 | | 24.2 | 24.2 |
| >64 | | 17.8 | 17.8 |
| Height (cm) | 0.4 | 162.4 (0.01) | 162.4 (0.01) |
| Body mass index (kg/m$^2$) | 0 | | |
| <18.5 | | 0.8 | 0.8 |
| 18.5–24.9 | | 38.7 | 38.7 |
| 25.0–29.9 | | 36.5 | 36.5 |
| >29.9 | | 24.1 | 24.1 |
| Non-white ethnicity | 0.5 | 5.4 | 5.4 |
| Education | 1.2 | | |
| College, university or other professional degree | | 36.9 | 37.3 |
| A levels, AS levels or equivalent | | 11.8 | 11.9 |
| O levels, GCSEs or equivalent | | 23.3 | 23.5 |
| CSE or equivalent | | 5.4 | 5.4 |
| NVQ, HND, HNC or equivalent | | 4.5 | 4.6 |
| Other | | 17.0 | 17.3 |
| Annual household income | 0.4 | | |
| <£18 000 | | 20.4 | 20.6 |
| £18,000-£30 999 | | 21.7 | 21.8 |
| £31,000-£51 999 | | 20.9 | 21.0 |
| £52,000-£100 000 | | 15.3 | 15.4 |
| >£100 000 | | 4.0 | 4.0 |
| Do not know/prefer not to answer | | 17.3 | 17.4 |
| Townsend deprivation index | 0.1 | −1.33 (0.01) | −1.33 (0.01) |
| Smoking history (pack-years) | 14.3 | | |
| None | | 59.3 | 69.1 |
| Up to 10 | | 7.9 | 9.2 |
| 11-20 | | 7.4 | 8.7 |
| 21–30 | | 5.2 | 6.1 |
| >30 | | 5.9 | 6.9 |
| Prevalent (baseline) COPD | 0 | 0.8 | 0.8 |
| Incident COPD hospitalisation/death | 0 | 0.4 | 0.4 |
| Spirometry* | 28.4 | | |
| FEV$_1$ z-score | | 0.00001 (0.002) | −0.0008 (0.002) |
| FVC z-score | | −0.000002 (0.002) | −0.001 (0.002) |
| FEV$_1$/FVC z-score | | 0.00003 (0.002) | 0.002 (0.002) |
| Asthma | 0 | 12.9 | 12.9 |
| Maternal history of COPD | 1.9 | | |
| No | | 90.0 | 91.7 |
| Yes | | 6.0 | 6.1 |
| Don't know | | 2.2 | 2.2 |
| Paternal history of COPD | 3.0 | | |
| No | | 81.7 | 84.2 |
| Yes | | 10.3 | 10.6 |
| Don't know | | 5.0 | 5.2 |

Continued

**Table 1** Continued

| Characteristic | % data imputed | Mean (SE) or % in observed data | Mean (SE) or % in imputed data |
|---|---|---|---|
| Age at menarche, years | 3.3 | | |
| <12 | | 19.4 | 20.0 |
| 12-15 | | 71.7 | 74.1 |
| >15 | | 5.7 | 5.9 |
| Menopause status | 16.1 | | |
| No | | 23.4 | 26.4 |
| Yes | | 60.5 | 73.6 |
| Age at natural menopause, years | 20.8 | | |
| <47 | | 11.4 | 20.7 |
| 47–49 | | 8.9 | 12.8 |
| 50–52 | | 19.7 | 23.7 |
| >52 | | 15.8 | 16.3 |
| Did not undergo natural menopause | | 23.4 | 26.4 |
| Parity | 1.9 | | |
| Nulliparous | | 14.7 | 15.0 |
| Parous | | 83.4 | 85.0 |
| 1 | | 11.9 | 12.2 |
| 2 | | 33.7 | 34.4 |
| 3 | | 24.8 | 25.3 |
| >3 | | 12.9 | 13.2 |
| PCOS/ovarian cysts | 0 | 1.7 | 1.7 |
| Endometriosis | 0 | 1.5 | 1.5 |
| Oral contraception use, years | 0.5 | | |
| Never | | 18.9 | 19.0 |
| Ever | | 80.6 | 81.0 |
| 1 | | 11.0 | 11.1 |
| 2-4 | | 18.6 | 18.7 |
| 5-9 | | 17.7 | 17.7 |
| 10-15 | | 17.4 | 17.4 |
| >15 | | 16.0 | 16.0 |
| Hormone replacement therapy, years | 5.7 | | |
| Never | | 61.4 | 63.7 |
| Ever | | 32.9 | 36.3 |
| 1-2 | | 10.7 | 11.8 |
| 3-5 | | 8.4 | 9.1 |
| 6-10 | | 5.7 | 6.2 |
| >10 | | 8.2 | 9.2 |
| Gynaecological surgery | 1.8 | | |
| No | | 80.4 | 81.7 |
| Hysterectomy | | 9.9 | 10.1 |
| Bilateral oophorectomy | | 0.4 | 0.4 |
| Both | | 7.6 | 7.8 |

*Spirometry measures were standardised by age, sex and height.

A levels, advanced levels; AS levels, advanced subsidiary levels; COPD, chronic obstructive pulmonary disease; CSE, certificate of secondary education; FEV$_1$, forced expiratory volume at 1 second; FVC, forced vital capacity; GCSE, General Certificate of Secondary Educations; HNC, higher national certificate; HND, higher national diploma; NVQ, national vocational qualifications; O levels, ordinary levels; PCOS, polycystic ovary syndrome; SE, standard error.

**Table 2** Cox regression analyses of female reproductive health indicators with incident COPD-related hospitalisation/death during follow-up in women without COPD at baseline (n=271 271)

| Reproductive health indicator | % of participants | Person-years of follow-up | Number of COPD events | Age-adjusted HR | 95% CI | Multiple-adjusted* HR | 95% CI |
|---|---|---|---|---|---|---|---|
| **Age at menarche** | | | | | | | |
| <12 years | 20.0 | 326 027 | 275 | 1.31 | 1.14 to 1.51 | 1.15 | 1.00 to 1.32 |
| 12–15 years | 74.1 | 1 210 146 | 768 | 1.0 | NA | 1.0 | NA |
| >15 years | 5.9 | 95 771 | 95 | 1.61 | 1.29 to 1.99 | 1.37 | 1.11 to 1.71 |
| **Menopause status** | | | | | | | |
| No | 26.6 | 437 890 | 97 | 1.0 | NA | 1.0 | NA |
| Yes | 73.4 | 1 194 055 | 1041 | 1.47 | 1.12 to 1.94 | 1.07 | 0.82 to 1.41 |
| **Age at natural menopause†** | | | | | | | |
| <47 years | 20.3 | 202 811 | 270 | 2.21 | 1.83 to 2.68 | 1.44 | 1.19 to 1.75 |
| 47–49 years | 15.8 | 158 718 | 140 | 1.55 | 1.24 to 1.94 | 1.25 | 1.00 to 1.57 |
| 50–52 years | 35.4 | 354 774 | 219 | 1.0 | NA | 1.0 | NA |
| >52 years | 28.4 | 283 721 | 161 | 0.86 | 0.69 to 1.06 | 0.93 | 0.75 to 1.15 |
| **Parity, binary** | | | | | | | |
| Nulliparous | 15.1 | 244 938 | 116 | 0.77 | 0.63 to 0.93 | 0.92 | 0.76 to 1.12 |
| Parous | 84.9 | 1 387 009 | 1022 | 1.0 | NA | 1.0 | NA |
| **Parity, ordered categories** | | | | | | | |
| 0 | 15.1 | 244 938 | 116 | 1.0 | NA | 1.0 | NA |
| 1 | 12.2 | 198 124 | 139 | 1.41 | 1.10 to 1.81 | 1.06 | 0.82 to 1.35 |
| 2 | 34.4 | 562 339 | 308 | 0.96 | 0.78 to 1.19 | 0.92 | 0.74 to 1.14 |
| 3 | 25.3 | 413 347 | 293 | 1.24 | 1.00 to 1.54 | 1.06 | 0.85 to 1.33 |
| >3 | 13.1 | 213 195 | 282 | 2.21 | 1.78 to 2.75 | 1.45 | 1.16 to 1.82 |
| **PCOS/ovarian cysts** | | | | | | | |
| No | 98.3 | 1 604 641 | 1108 | 1.0 | NA | 1.0 | NA |
| Yes | 1.7 | 27 304 | 30 | 1.61 | 1.12 to 2.32 | 1.61 | 1.12 to 2.32 |
| **Endometriosis** | | | | | | | |
| No | 98.5 | 1 607 817 | 1127 | 1.0 | NA | 1.0 | NA |
| Yes | 1.5 | 24 131 | 11 | 0.79 | 0.44 to 1.43 | 0.85 | 0.47 to 1.53 |
| **Oral contraception** | | | | | | | |
| Never used | 18.9 | 308 641 | 295 | 1.0 | NA | 1.0 | NA |
| Ever used | 81.1 | 1 323 303 | 843 | 0.95 | 0.83 to 1.09 | 0.85 | 0.74 to 0.97 |
| **Years of oral contraception use** | | | | | | | |
| 0 | 18.9 | 308 641 | 295 | 1.0 | NA | 1.0 | NA |
| 1 | 11.1 | 180 195 | 137 | 1.05 | 0.85 to 1.29 | 0.84 | 0.69 to 1.04 |
| 2–4 | 18.7 | 306 053 | 246 | 1.05 | 0.88 to 1.24 | 0.95 | 0.80 to 1.13 |
| 5–9 | 17.8 | 290 554 | 181 | 0.92 | 0.76 to 1.11 | 0.86 | 0.71 to 1.05 |
| 10–15 | 17.5 | 285 766 | 153 | 0.84 | 0.69 to 1.03 | 0.74 | 0.61 to 0.91 |
| >15 | 16.1 | 260 734 | 126 | 0.83 | 0.67 to 1.04 | 0.75 | 0.60 to 0.93 |
| **HRT** | | | | | | | |
| Never used | 63.9 | 1 042 083 | 510 | 1.0 | NA | 1.0 | NA |
| Ever used | 36.1 | 589 861 | 628 | 1.49 | 1.31 to 1.69 | 1.15 | 1.01 to 1.30 |
| **Years of HRT** | | | | | | | |
| 0 | 63.9 | 1 042 083 | 510 | 1.0 | NA | 1.0 | NA |
| 1–2 | 11.7 | 190 902 | 240 | 1.83 | 1.54 to 2.16 | 1.33 | 1.12 to 1.57 |
| 3–5 | 9.1 | 147 962 | 162 | 1.63 | 1.34 to 1.97 | 1.17 | 0.97 to 1.42 |
| 6–10 | 6.2 | 100 921 | 74 | 1.0 | 0.77 to 1.31 | 0.87 | 0.66 to 1.14 |

Continued

**Table 2** Continued

| Reproductive health indicator | % of participants | Person-years of follow-up | Number of COPD events | Age-adjusted | | Multiple-adjusted* | |
|---|---|---|---|---|---|---|---|
| | | | | HR | 95% CI | HR | 95% CI |
| >10 | 9.2 | 150 077 | 152 | 1.27 | 1.04 to 1.55 | 1.06 | 0.87 to 1.29 |
| Gynaecological surgery | | | | | | | |
| No | 81.8 | 1 335 512 | 765 | 1.0 | NA | 1.0 | NA |
| Hysterectomy | 10.1 | 164 869 | 213 | 1.77 | 1.51 to 2.07 | 1.49 | 1.28 to 1.74 |
| Bilateral oophorectomy | 0.4 | 5957 | 6 | 1.56 | 0.70 to 3.48 | 1.36 | 0.61 to 3.04 |
| Both | 7.8 | 125 607 | 154 | 1.68 | 1.41 to 2.00 | 1.42 | 1.19 to 1.69 |

*Multiple-adjusted for age, height, body mass index (log-transformed), ethnicity, education, household income, Townsend deprivation index, smoking history, maternal COPD and paternal COPD.
†Age at menopause was analysed only among women who experienced natural menopause before baseline (n=159 571).
COPD, chronic obstructive pulmonary disease; HRT, hormone replacement therapy; PCOS, polycystic ovary syndrome.

remaining associations were unchanged from the main analyses.

We also investigated associations of reproductive health indicators with risk of COPD-related mortality, and not hospitalisation, among women of UK Biobank. The associations from these analyses of 273 441 women (including those with COPD reported at baseline) were null for all reproductive health indicators (online supplementary file table S9), except age at menarche <12 years compared with 12 to 15 years (HR 1.52; 95% CI: 1.08 to 2.13). However, there were only 179 COPD-related deaths in UK Biobank and many categories of our reproductive health indicators experienced fewer than 10 events. Complete-case analyses of the risk of COPD hospitalisation/death (online supplementary file Table S10) and lung function (online supplementary file table 11) also produced similar results to those from our imputed data set.

## DISCUSSION

In this large, prospective study, several female reproductive health indicators were associated with COPD hospitalisation/death and/or lung function. Parity greater than three was associated with decreased baseline $FEV_1$/FVC, a marker of airway obstruction, and increased risk of hospitalisation/death from COPD during follow-up. Ever using OC was associated with greater $FEV_1$/FVC and lower risk of COPD hospitalisation/death. Associations of hysterectomy, HRT use and a history of PCOS or ovarian cysts showed inconsistent findings for $FEV_1$/FVC and COPD hospitalisation/death. This inconsistency may be explained by the fact that spirometry was measured in participants at baseline (age 40 to 69) and $FEV_1$ and $FEV_1$/FVC may have declined during follow-up. Conversely, many patients diagnosed with stage 1 COPD do not go on to develop COPD symptoms or COPD resulting in hospitalisation/death.[27] There is also high intra-individual variability in $FEV_1$.[28] Spirometry assessment at one time point may therefore not be as accurate at identifying clinically-relevant COPD compared with register data on hospitalisation/death.

A study of 424 797 postmenopausal women in the United States reported no difference in COPD mortality by parity.[9] In contrast, our present study reports higher risk of COPD hospitalisation/death among women with more than three births compared with nulliparous women. However, we investigated both COPD-related hospitalisation and death, and we were therefore able to capture less severe COPD cases. A study of 70 965 women in the Nurses' Health Study reported no evidence of an association between HRT ever use and COPD incidence, although the investigators included both probable (self-reported physician diagnosis along with a record of spirometry or imaging at the time of diagnosis or COPD documented on a death register) and definite (self-reported physician diagnosis and $FEV_1$ <80% predicted) cases of COPD.[8] Here we report a positive association between HRT use and COPD hospitalisation/death. This is the first study to assess COPD risk with regards to age at menarche, menopause status, hysterectomy, gynaecological surgery, endometriosis or PCOS.

Two previous studies reported reduced $FEV_1$ in adult women who experienced menarche before age 10[10 12] but neither found evidence of an association between age at menarche and $FEV_1$/FVC. A Mendelian randomisation study of adolescent and adult women across multiple European cohorts, including a subset from UK Biobank, estimated a 24.8 mL increase in FVC per year increase in age at menarche among adult women and a −56.6 mL decrease in FVC per year increase in age at menarche among adolescent girls.[11] This study reported no association between age at menarche and $FEV_1$/FVC.[11] Notably, this Mendelian randomisation study did not explore non-linear associations between age at menarche and measures of lung function. We observed lower $FEV_1$ and FVC in adult women with late menarche, and lower FVC and $FEV_1$/FVC in adult women with early menarche.

A recent systematic review, which included studies from UK Biobank,[16] reported that postmenopausal status was associated with reduced $FEV_1$ and FVC but not $FEV_1$/FVC.[18] Current HRT use has been associated with both higher[14] and lower[15] lung function. One study reported

**Table 3** Linear regression analyses of female reproductive health indicators with baseline spirometry measures (n=273 441)

| Reproductive health indicator | Mean change in $FEV_1$ z-score (95% CI) | | Mean change in FVC z-score (95% CI) | | Mean change in $FEV_1$/FVC z-score (95% CI) | |
|---|---|---|---|---|---|---|
| | Unadjusted | Multiple-adjusted* | Unadjusted | Multiple-adjusted* | Unadjusted | Multiple-adjusted* |
| **Age at menarche** | | | | | | |
| <12 years | −0.04 (−0.05 to −0.03) | −0.01 (−0.02 to 0.00) | −0.05 (−0.06 to −0.04) | −0.01 (−0.02 to −0.01) | 0.02 (0.01 to 0.03) | −0.02 (−0.03 to −0.01) |
| 12–15 years | 0 | 0 | 0 | 0 | 0 | 0 |
| >15 years | −0.08 (−0.10 to −0.06) | −0.03 (−0.05 to −0.01) | −0.06 (−0.08 to −0.04) | −0.03 (−0.05 to −0.02) | −0.05 (−0.07 to −0.02) | 0.00 (−0.02 to 0.02) |
| **Menopause status** | | | | | | |
| No | 0 | 0 | 0 | 0 | 0 | 0 |
| Yes | −0.03 (−0.04 to −0.02) | −0.04 (−0.05 to −0.02) | −0.03 (−0.04 to −0.02) | −0.04 (−0.06 to −0.03) | 0.00 (−0.01 to 0.01) | 0.01 (−0.01 to 0.02) |
| **Age at natural menopause†** | | | | | | |
| <47 years | −0.11 (−0.12 to −0.09) | −0.04 (−0.05 to −0.02) | −0.09 (−0.11 to −0.08) | −0.04 (−0.05 to −0.02) | −0.05 (−0.07 to −0.04) | −0.01 (−0.02 to 0.01) |
| 47–49 years | −0.05 (−0.07 to −0.04) | −0.02 (−0.04 to 0.00) | −0.04 (−0.06 to −0.03) | −0.02 (−0.03 to 0.00) | −0.03 (−0.05 to −0.02) | −0.01 (−0.03 to 0.01) |
| 50–52 years | 0 | 0 | 0 | 0 | 0 | 0 |
| >52 years | 0.01 (0.00 to 0.03) | 0.00 (−0.01 to 0.02) | 0.00 (−0.02 to 0.01) | 0.00 (−0.02 to 0.01) | 0.03 (0.02 to 0.05) | 0.01 (0.00 to 0.02) |
| **Parity, binary** | | | | | | |
| Nulliparous | −0.02 (−0.03 to −0.01) | −0.04 (−0.06 to −0.03) | −0.04 (−0.05 to −0.03) | −0.07 (−0.08 to −0.05) | 0.04 (0.03 to 0.05) | 0.04 (0.03 to 0.05) |
| Parous | 0 | 0 | 0 | 0 | 0 | 0 |
| **Parity, ordered categories** | | | | | | |
| 0 | 0 | 0 | 0 | 0 | 0 | 0 |
| 1 | −0.02 (−0.04 to −0.01) | 0.01 (0.00 to 0.03) | −0.01 (−0.02 to 0.01) | 0.02 (0.01 to 0.04) | −0.04 (−0.06 to −0.03) | −0.02 (−0.04 to 0.00) |
| 2 | 0.04 (0.03 to 0.06) | 0.04 (0.03 to 0.06) | 0.06 (0.05 to 0.08) | 0.07 (0.06 to 0.08) | −0.03 (−0.05 to −0.02) | −0.04 (−0.06 to −0.03) |
| 3 | 0.03 (0.02 to 0.05) | 0.05 (0.04 to 0.07) | 0.05 (0.04 to 0.07) | 0.08 (0.06 to 0.09) | −0.04 (−0.06 to −0.03) | −0.04 (−0.06 to −0.03) |
| >3 | −0.03 (−0.05 to −0.01) | 0.05 (0.04 to 0.07) | −0.01 (−0.02 to 0.01) | 0.08 (0.07 to 0.10) | −0.07 (−0.08 to −0.05) | −0.06 (−0.07 to −0.04) |
| **PCOS/ovarian cysts** | | | | | | |
| No | 0 | 0 | 0 | 0 | 0 | 0 |
| Yes | −0.02 (−0.05 to 0.02) | −0.01 (−0.04 to 0.03) | −0.05 (−0.08 to −0.01) | −0.03 (−0.07 to 0.00) | 0.06 (0.03 to 0.09) | 0.05 (0.02 to 0.09) |
| **Endometriosis** | | | | | | |
| No | 0 | 0 | 0 | 0 | 0 | 0 |
| Yes | 0.07 (0.05 to 0.08) | 0.03 (0.01 to 0.04) | 0.08 (0.07 to 0.09) | 0.02 (0.01 to 0.03) | 0.06 (0.02 to 0.10) | 0.07 (0.03 to 0.10) |
| **Oral contraception** | | | | | | |
| Never used | 0 | 0 | 0 | 0 | 0 | 0 |
| Ever used | 0.01 (−0.03 to 0.04) | 0.01 (−0.02 to 0.05) | −0.02 (−0.06 to 0.01) | −0.02 (−0.05 to 0.01) | −0.01 (−0.03 to 0.00) | 0.01 (0.003 to 0.03) |
| **Years of oral contraception** | | | | | | |
| 0 | 0 | 0 | 0 | 0 | 0 | 0 |
| 1 | 0.03 (0.01 to 0.04) | 0.01 (−0.01 to 0.03) | 0.04 (0.02 to 0.06) | 0.01 (−0.01 to 0.03) | −0.03 (−0.04 to −0.01) | 0.01 (−0.004 to 0.03) |

Continued

**Table 3** Continued

| Reproductive health indicator | Mean change in FEV$_1$ z-score (95% CI) | | Mean change in FVC z-score (95% CI) | | Mean change in FEV$_1$/FVC z-score (95% CI) | |
|---|---|---|---|---|---|---|
| | Unadjusted | Multiple-adjusted* | Unadjusted | Multiple-adjusted* | Unadjusted | Multiple-adjusted* |
| 2–4 | 0.05 (0.04 to 0.07) | 0.02 (0.01 to 0.04) | 0.06 (0.05 to 0.08) | 0.02 (0.01 to 0.03) | 0.00 (−0.02 to 0.01) | 0.02 (0.003 to 0.03) |
| 5–9 | 0.08 (0.07 to 0.10) | 0.03 (0.02 to 0.05) | 0.09 (0.08 to 0.10) | 0.03 (0.01 to 0.04) | 0.00 (−0.02 to 0.01) | 0.02 (0.003 to 0.03) |
| 10–15 | 0.08 (0.07 to 0.10) | 0.04 (0.02 to 0.05) | 0.10 (0.08 to 0.11) | 0.04 (0.02 to 0.05) | −0.02 (−0.03 to 0.00) | 0.01 (−0.002 to 0.03) |
| >15 | 0.07 (0.05 to 0.08) | 0.02 (0.01 to 0.04) | 0.08 (0.07 to 0.10) | 0.02 (0.00, 0.03) | −0.03 (−0.04 to −0.01) | 0.01 (−0.002 to 0.03) |
| HRT | | | | | | |
| Never used | 0 | 0 | 0 | 0 | 0 | 0 |
| Ever used | −0.02 (−0.02 to −0.01) | 0.00 (0.00 to 0.01) | −0.01 (−0.01 to 0.00) | 0.00 (−0.01 to 0.01) | −0.02 (−0.03 to −0.01) | 0.02 (0.01 to 0.03) |
| Years of HRT | | | | | | |
| 0 | 0 | 0 | 0 | 0 | 0 | 0 |
| 1–2 | −0.03 (−0.04 to −0.01) | 0.00 (−0.01 to 0.02) | −0.01 (−0.03 to 0.00) | 0.00 (−0.02 to 0.01) | −0.03 (−0.05 to −0.02) | 0.02 (0.01 to 0.04) |
| 3–5 | −0.03 (−0.04 to −0.01) | 0.00 (−0.01 to 0.02) | −0.02 (−0.03 to 0.00) | 0.00 (−0.02 to 0.02) | −0.02 (−0.04 to 0.00) | 0.02 (0.00 to 0.03) |
| 6–10 | 0.00 (−0.02 to 0.02) | 0.01 (−0.01 to 0.02) | 0.00 (−0.02 to 0.02) | 0.00 (−0.02 to 0.02) | 0.00 (−0.02 to 0.01) | 0.02 (0.00 to 0.04) |
| >10 | 0.00 (−0.02 to 0.01) | 0.00 (−0.01 to 0.02) | 0.01 (0.00 to 0.03) | 0.00 (−0.01 to 0.02) | −0.03 (−0.05 to −0.01) | 0.01 (−0.01 to 0.03) |
| Gynaecological surgery | | | | | | |
| No | 0 | 0 | 0 | 0 | 0 | 0 |
| Hysterectomy | −0.03 (−0.05 to −0.02) | 0.01 (0.00 to 0.03) | −0.05 (−0.06 to −0.04) | 0.00 (−0.02 to 0.01) | 0.03 (0.01 to 0.04) | 0.04 (0.02 to 0.06) |
| Bilateral oophorectomy | −0.05 (−0.12 to 0.02) | −0.02 (−0.08 to 0.05) | −0.07 (−0.14 to 0.00) | −0.04 (−0.11 to 0.03) | 0.04 (−0.03 to 0.12) | 0.05 (−0.02 to 0.12) |
| Both | −0.04 (−0.05 to −0.02) | 0.00 (−0.01 to 0.02) | −0.07 (−0.08 to −0.05) | −0.02 (−0.04 to −0.01) | 0.05 (0.04 to 0.07) | 0.05 (0.04 to 0.07) |

*Adjusted for age, body mass index (log-transformed), ethnicity, education, household income, Townsend deprivation index, smoking history, asthma, maternal COPD and paternal COPD.
†Age at menopause was analysed only among women who experienced natural menopause before baseline (n=161 069).
FEV$_1$, forced expiratory volume in 1 second; FVC, forced vital capacity; HRT, hormone replacement therapy; PCOS, polycystic ovary syndrome.

**Table 4** Direction of association for female reproductive health indicators with risk of COPD-related hospitalisation/death and spirometry measures

| Reproductive health indicator | COPD-related hospitalisation/death | FEV$_1$ | FVC | FEV$_1$/FVC |
|---|---|---|---|---|
| Age at menarche | | | | |
| <12 years | = | = | ↓ | ↓ |
| 12–15 years | NA | NA | NA | NA |
| >15 years | ↑ | ↓ | ↓ | = |
| Menopause status | | | | |
| No | NA | NA | NA | NA |
| Yes | = | ↓ | ↓ | = |
| Age at natural menopause | | | | |
| <47 years | ↑ | ↓ | ↓ | = |
| 47–49 years | = | = | = | = |
| 50–52 years | NA | NA | NA | NA |
| >52 years | = | = | = | = |
| Parity, binary | | | | |
| Nulliparous | = | ↓ | ↓ | ↑ |
| Parous | NA | NA | NA | NA |
| Parity, ordered categories | | | | |
| 0 | NA | NA | NA | NA |
| 1 | = | = | ↑ | ↓ |
| 2 | = | ↑ | ↑ | ↓ |
| 3 | = | ↑ | ↑ | ↓ |
| >3 | ↑ | ↑ | ↑ | ↓ |
| PCOS/ovarian cysts | | | | |
| No | NA | NA | NA | NA |
| Yes | ↑ | = | = | ↑ |
| Endometriosis | | | | |
| No | NA | NA | NA | NA |
| Yes | = | = | = | ↑ |
| Oral contraception | | | | |
| Never used | NA | NA | NA | NA |
| Ever used | ↓ | ↑ | ↑ | = |
| Years of oral contraception | | | | |
| 0 | NA | NA | NA | NA |
| 1 | = | = | = | = |
| 2-4 | = | ↑ | ↑ | ↑ |
| 5-9 | = | ↑ | ↑ | ↑ |
| 10-15 | ↓ | ↑ | ↑ | = |
| >15 | ↓ | ↑ | = | = |
| Hormone replacement therapy | | | | |
| Never used | NA | NA | NA | NA |
| Ever used | ↑ | = | = | ↑ |
| Years of hormone replacement therapy | | | | |
| 0 | NA | NA | NA | NA |
| 1-2 | ↑ | = | = | ↑ |

Continued

| Table 4 | Continued | | | |
|---|---|---|---|---|
| Reproductive health indicator | COPD-related hospitalisation/death | FEV$_1$ | FVC | FEV$_1$/FVC |
| 3-5 | = | = | = | = |
| 6-10 | = | = | = | = |
| >10 | = | = | = | = |
| Gynaecological surgery | | | | |
| No | NA | NA | NA | NA |
| Hysterectomy | ↑ | = | = | ↑ |
| Bilateral oophorectomy | = | = | = | = |
| Both | ↑ | = | ↓ | ↑ |

COPD, chronic obstructive pulmonary disease; FEV$_1$, forced expiratory volume in 1 second; FVC, forced vital capacity; NA, not applicable where the category is the reference group; PCOS, polycystic ovary syndrome.

an association between HRT use and greater FEV$_1$/FVC.[14] Others have identified differences in lung function according to continuous/cyclic and combined/oestrogen-only HRT.[20 21] Based on the information from participants in UK Biobank, we could not stratify by HRT subtypes.

There are sex differences in lung anatomy and physiology throughout the life course.[1 2 4] During gestation, female foetuses have fewer bronchi, smaller lungs and produce surfactant earlier than male foetuses.[1] Female neonates and children have greater lung function and faster lung growth than their male counterparts, and boys are more likely to be diagnosed with asthma during childhood than girls, while girls are twice as likely as boys to be diagnosed with asthma after puberty.[1] These findings suggest that female sex steroids can accelerate lung maturation and termination of lung growth, which may partially explain why early menarche associates with lower FEV$_1$/FVC in adulthood. During the reproductive years, parity and PCOS relate to an increase in circulating oestrogen, which has been implicated in the pathogenesis of inflammatory lung diseases including COPD.[6 29–31] Although OC use appears to be protective against COPD, some researchers have suggested that OC use attenuates the fluctuations in sex steroids, which correlates with fluctuations in endothelial nitric oxide, a potent bronchodilator.[1] Additionally, exogenous hormones may not be as influential during reproductive years but may become more influential in after menopause, when endogenous hormone production is low.[6] This is supported by our findings of opposing associations with COPD risk for OC use (lower risk) during reproductive period and HRT use (higher risk) after menopause.

The effect of female sex hormones on COPD and lung health may vary depending on the net effects on airway smooth muscle, inflammation and cigarette smoke metabolism. Oestrogens can regulate the synthesis of nitric oxide, a potent bronchodilator, within human bronchial epithelium,[1 32] as well as regulate airway bronchoconstriction in response to acetylcholine or histamine.[33 34] There is also evidence that progesterone may affect airway smooth muscle constriction, although these reports are limited to animal studies.[35] In addition to regulating bronchodilation and bronchoconstriction, in vitro studies report that oestrogen can also modulate the proliferation of airway smooth muscle cells,[36 37] which is a key feature of COPD.[38] Oestrogen's proinflammatory and anti-inflammatory effects have been documented in other female-predominant diseases[39] and may also play a role in COPD, as chronic inflammation is another hallmark of the disease. Finally, oestrogen is an important regulator of toxic cigarette smoke metabolites through cytochrome P450 enzymes.[1] In a mouse model of COPD, Glassberg *et al* reported that oestradiol administration after ovariectomy protected against cigarette smoke-induced alveolar septal destruction, macrophage infiltration and decreased other markers of inflammation compared with ovariectomy without oestradiol administration.[40] In the same mouse model, however, non-ovariectomised female mice also had increased small airway remodelling following cigarette smoke exposure when compared with both male mice and ovariectomised female mice.[41] These timing-dependent, dose-dependent and cell-dependent signalling pathways may explain why exposure to female sex steroids, and oestrogen in particular, have potentially contradictory effects on respiratory health.

We cannot exclude a potential influence of selection bias, due to the low participation rate in UK Biobank. Women in the cohort are less likely to have ever smoked, have greater educational attainment, but a similar number of offspring, as compared with the general UK population (https://www.ons.gov.uk/). There is also a lower prevalence of self-reported COPD in UK Biobank (0.8%) compared with physician-diagnosed COPD in the general female UK population (1.68%).[42] Overall, women participating in UK Biobank reflect a higher-educated, more health-conscious group, with more modest underlying risk for COPD compared with the general population. We can only speculate as to how these selection factors might have influenced our associations of interest, but we propose that the most likely direction of any selection bias is towards the null (resulting in an

 Tang R, *et al. BMJ Open* 2019;**9**:e030318. doi:10.1136/bmjopen-2019-030318

underestimation). Strengths of this study include the large, prospective design. However, we relied on participants to self-report reproductive history which might have led to measurement error. Years of OC or HRT use were estimated as the difference between ages at first and last use. This likely resulted in overestimation of the duration of use, which might have attenuated our findings. In addition, we were unable to distinguish between reversible and non-reversible airway obstruction at baseline, as a bronchodilator was not administered before spirometry was conducted. Although we adjusted for self-reported asthma at baseline, this may have led to overestimation of COPD-related airway obstruction among participants with undiagnosed asthma. Lastly, we were unable to adjust for environmental tobacco smoke exposure as these data were not collected in UK Biobank. We therefore cannot exclude the possibility of residual bias from environmental tobacco smoke.

In conclusion, our findings indicate that several female reproductive health indicators across the life course are associated with risk of COPD hospitalisation/death and add to the indirect evidence suggesting that female reproductive hormones may have a role in COPD pathogenesis. Further studies are necessary to understand the opposing associations of hysterectomy, HRT and PCOS/ovarian cysts with COPD and objective measures of airway obstruction.

**Author affiliations**
[1]Bristol Medical School, Faculty of Health Sciences, University of Bristol, Bristol, UK
[2]Keenan Research Centre for Biomedical Science, St Michael's Hospital, Toronto, Ontario, Canada
[3]MRC Integrative Epidemiology Unit, University of Bristol, Bristol, UK
[4]Department of Population Health Sciences, Bristol Medical School, University of Bristol, Bristol, UK
[5]Centre for Fertility and Health, Norwegian Institute of Public Health, Oslo, Norway

**Acknowledgements** This research has been conducted using the UK Biobank Resource under Application Number 6326. We thank all the participants and researchers involved.

**Contributors** All authors provided substantial contributions to the design of the work. RT analysed the data and drafted the manuscript. MCM and AF supervised the study, provided further data interpretation and revised the manuscript. All authors approved the final version of the manuscript and agree to be accountable for all aspects of the work in ensuring that questions related to the accuracy or integrity of any part of the work are appropriately investigated and resolved.

**Funding** This work was supported by the UK Medical Research Council grant MC/UU/12013/5, which provides infrastructure funding to the MRC Integrative Epidemiology Unit at the University of Bristol, and grant MR/M009351/1 as a fellowship to AF. This work was partly supported by the Research Council of Norway's Centres of Excellence scheme grant 262700.

**Competing interests** None declared.

**Patient consent for publication** Not required.

**Ethics approval** Data collection in UK Biobank was approved by the National Health Service National Research Ethics Service (Ref 11/NW/0382).

**Provenance and peer review** Not commissioned; externally peer reviewed.

**Data availability statement** Data are available in a public, open access repository.

**ORCID iD**
Rosalind Tang http://orcid.org/0000-0002-4379-6673

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
