## [Reviewer comments · BMJ Open]

ARTICLE DETAILS

TITLE (PROVISIONAL)	Female reproductive history in relation to chronic obstructive pulmonary disease and lung function in UK Biobank: a prospective population-based cohort study
AUTHORS	Tang, Rosalind; Fraser, Abigail; Magnus, Maria

VERSION 1 – REVIEW

REVIEWER	Anthony B Miller Dalla Lana School of Public health, University of Toronto, Canada
REVIEW RETURNED	17-Mar-2019

GENERAL COMMENTS	The authors have studied a cohort which, as they state, suffers from ascertainment bias. Thus, the findings can not be regarded as representative of those that might be derived from a general population sample. Nevertheless, they are probably internally valid. A major causal risk factor for COPD is smoking. Therefore, all associations derived from a study of the data derived from any cohort must be fully adjusted for potential confounding variables, critically cigarette smoking. Unfortunately, the authors have chosen to characterise cigarette smoking in terms of pack-years. This derived variable from both intensity and duration cannot be relied upon to fully adjust for smoking. Instead, the authors need to adjust fully both for intensity and duration of smoking independently, the latter, as Peto and his colleagues have demonstrated, being critical. Therefore the analysis must be repeated with full adjustment for smoking both by intensity and duration.
---

REVIEWER	Om P. Kurmi McMaster University, Canada
REVIEW RETURNED	24-Apr-2019

GENERAL COMMENTS	The paper by Tang et al have analyzed UK Biobank dataset to report on the association between female reproductive history with their lung function and also COPD hospitalization and mortality. This is an important area of research and under explored. The findings reported so far are inconsistent and therefore, the study will help to add to the existing literature. Despite being a large study and as also highlighted by the authors, this study is not representative of the UK population although they have collected data from many centres. The paper is well written and my comments/queries are as follows: 1. Initially the authors analysed the data after excluding self-reported COPD at baseline but later on excluded those who had
---

	airflow obstruction based on the spirometry during sensitivity analysis. It is not clear if during the sensitivity analysis, the authors excluded all those who had self-reported COPD and spirometry based airflow obstruction. Have you excluded other self-reported respiratory conditions such as asthma, TB, pneumonia at baseline. Excluding this give a much cleaner dataset and reduce chances of reverse causality. In addition to excluding baseline self-reported respiratory conditions, I will also adjust the main analysis to other co-morbidity such as cardiovascular and diabetes. 2. Smoking is a main risk factor for COPD. Although women are less likely to smoke but I would concentrate my analysis in never smokers only and at the same time adjust for ETS. Adjusting for smoking history are likely to have residual bias. 3. In one of the analysis the study compares lung function and incidence of COPD hospitalization/mortality in post-menopausal women with pre-menopausal women (reference). We know that pre-menopausal women are younger compared to post-menopausal women. I am not sure that just by adjusting for the age is enough? If there is enough sample, I would suggest you to analyze the post- versus pre-menopausal in a very narrow age-band. Alternatively, you could analyse the data for men and women separately and see if women of same age group as men have higher incidence of hospitalization and mortality. 4. Why do you combine the COPD hospitalization and mortality in one analysis. They are likely to have different grade of COPD and I would separate the analysis if possible. 5. You have reported that early or late menarche are associated with increased risk of COPD-related hospitalization /mortality and same for post-menopause. I was wondering if you can create a score using all the potential factors that you have identified and analyze the result based on it. 6. Your lung function indices findings for different reproductive history might have the same bias due to age. How about analyzing for percent predicted values, this should take account for age and height and also ethnicity.
--	---

REVIEWER	Pentti Nieminen Medical Informatics and Data Analysis Research Group, University of Oulu, Finland
REVIEW RETURNED	16-Jun-2019

GENERAL COMMENTS	I enjoyed reading this manuscript. I have reviewed this manuscript with a particular emphasis on the statistical methods and analyses used. It was particularly interesting to see how good the statistical reporting was. I have some observations that should receive the attention of the authors. 1. The statistical intensity of this manuscript was close to the average of articles published in general medical journals. However, the quality of statistical reporting is very high: 10 in a scale from 0 (poor) to 10 (very high). 2. Page 6, line 60: Please consider adding a reference to Schoenfeld residuals. 3. Page 7, line 32: I don't understand why you have analysed z-scores of spirometry measurements. I find z-score difficult to interpret compared to original values that have a clear clinical interpretation. Please help your readers and clarify. 4. Table 2: Consider adding the method (Cox regression) to the title. 5. Table 2, page 26, line14: Clarify what is meant with "ordinal (5)".
--

	6. Table 3: Consider adding the method (linear regression) to the title. 7. Tables 2: Is the total number of analysed subjects 271271 (in the title) or 159571 (footnote b)? Clarify whether you have used complete cases or imputed data in Tables 2 and 3.
--	--

VERSION 1 – AUTHOR RESPONSE

Reviewer(s)' Comments to Author:

Reviewer: 1

Reviewer Name: Anthony B Miller

Institution and Country: Dalla Lana School of Public health, University of Toronto, Canada

Please state any competing interests or state 'None declared': None

Please leave your comments for the authors below

The authors have studied a cohort which, as they state, suffers from ascertainment bias. Thus, the findings can not be regarded as representative of those that might be derived from a general population sample. Nevertheless, they are probably internally valid.

Comment 1: A major causal risk factor for COPD is smoking. Therefore, all associations derived from a study of the data derived from any cohort must be fully adjusted for potential confounding variables, critically cigarette smoking. Unfortunately, the authors have chosen to characterise cigarette smoking in terms of pack-years. This derived variable from both intensity and duration cannot be relied upon to fully adjust for smoking. Instead, the authors need to adjust fully both for intensity and duration of smoking independently, the latter, as Peto and his colleagues have demonstrated, being critical. Therefore the analysis must be repeated with full adjustment for smoking both by intensity and duration.

Response 1: We agree with Reviewer 1 that appropriate adjustment for cigarette smoke exposure is crucial for a study of COPD. In response to your comment, we have now conducted sensitivity analyses for COPD hospitalisation/death, as well as for spirometry measurements, adjusting for smoking history as two covariates (history in years and average number of cigarettes per day) rather than smoking history in pack-years. We report the findings adjusting the associations of female reproductive health indicators with COPD hospitalisation/death and lung function for both duration of smoking and smoking intensity in online supplementary Tables S4 and S7. These findings were similar to those in the main analysis adjusting for pack years of smoking in Tables 2-3. Given the similar magnitude and direction of associations, we decided to keep smoking history adjusted as a single covariate (pack-years) in the Main Document to allow comparison of our findings with previous studies of reproductive history and COPD.¹

Results (p. 11)

“We report the results of our sensitivity analyses, after 1) further adjusting for baseline smoking status; 2) adjusting for smoking history as two covariates including number of years of smoking and average number of cigarettes per day; 3) excluding women who have ever smoked; or 4) further adjusting for cardiovascular disease or diabetes reported at baseline, in the online supplementary file (Tables S4-S8).”

Reviewer: 2

Reviewer Name: Om P. Kurmi

Institution and Country: McMaster University, Canada

Please state any competing interests or state 'None declared': None

Please leave your comments for the authors below

The paper by Tang et al have analyzed UK Biobank dataset to report on the association between female reproductive history with their lung function and also COPD hospitalization and mortality. This is an important area of research and under explored. The findings reported so far are inconsistent and therefore, the study will help to add to the existing literature. Despite being a large study and as also

highlighted by the authors, this study is not representative of the UK population although they have collected data from many centres. The paper is well written and my comments/queries are as follows:

Comment 1: Initially the authors analysed the data after excluding self-reported COPD at baseline but later on excluded those who had airflow obstruction based on the spirometry during sensitivity analysis. It is not clear if during the sensitivity analysis, the authors excluded all those who had self-reported COPD and spirometry based airflow obstruction. Have you excluded other self-reported respiratory conditions such as asthma, TB, pneumonia at baseline. Excluding this give a much cleaner dataset and reduce chances of reverse causality. In addition to excluding baseline self-reported respiratory conditions, I will also adjust the main analysis to other co-morbidity such as cardiovascular and diabetes.

Response 1: Thank you for identifying this issue in clarity. We had excluded participants with self-reported COPD from our sensitivity analyses as well as from the main analyses, but we now see that we had not clearly documented this in the text. In response to your comment, we have revised the text and further excluded participants who reported asthma, tuberculosis or pneumonia at baseline from our sensitivity analyses. Of the 217 512 participants without respiratory illness(es) reported at baseline, 392 participants experienced COPD-related hospitalisation/death. The findings from this smaller sample (online supplementary file Table S3) were similar to those in our main analyses (Table 2), with the two exceptions. Parity greater than three compared to zero had a null association with COPD. For years of oral contraception use, one year or five to nine years of use compared to zero was associated with lower risk of COPD-related hospitalisation/death (online supplementary file Table S3). These findings with regards to contraception are consistent with ever use compared to never use and 10 or more years of contraception compared to zero both being associated with lower risk of COPD-related hospitalisation/death.

Methods (p. 7)

"We performed sensitivity analyses of[...] 3) all reproductive health indicators after excluding participants with respiratory conditions (i.e. COPD, asthma, tuberculosis or pneumonia) reported at baseline or $FEV_1/FVC < 0.7$ at baseline, as they possibly had undiagnosed COPD."

Results (p. 11)

"55 929 women had reported either a respiratory illness (asthma, COPD, tuberculosis and/or pneumonia) and/or $FEV_1/FVC < 0.7$ at baseline. However, our findings after excluding these women from our analyses of COPD-related hospitalisation/death were similar to our main findings (online supplementary file Table S3)."

In response to your comment, we have also conducted sensitivity analyses adjusting for cardiovascular disease and diabetes reported at baseline. These results are reported in online supplementary file Tables S6 and S8. Given the similarity in our findings after further adjustment for these comorbidities, we decided to present these further adjusted analyses in the online supplementary file.

Results (p. 11)

"We report the results of our sensitivity analyses, after 1) further adjusting for baseline smoking status; 2) adjusting for smoking history as two covariates including duration in years of smoking and average number of cigarettes per day; 3) excluding women who have ever smoked; or 4) further adjusting for cardiovascular disease or diabetes reported at baseline, in the online supplementary file (Tables S4-S8). The direction and magnitude of associations remained similar for risk of COPD hospitalisation/death (online supplementary file Tables S4-S6) and change in spirometry measures (online supplementary file Tables S7 and S8) after further adjustment for the above covariates."

Comment 2: Smoking is a main risk factor for COPD. Although women are less likely to smoke but I would concentrate my analysis in never smokers only and at the same time adjust for ETS. Adjusting for smoking history are likely to have residual bias.

Response 2: We have now completed analyses including only women who have never smoked (n=161 626) and the findings are reported in online supplementary file Table S5. The following associations with COPD death/hospitalisation remained after excluding women who have ever smoked: older age at menarche, >15 years of oral contraception use, 1-2 years of hormone

replacement therapy, hysterectomy and both hysterectomy/bilateral oophorectomy. Early age at menarche was also positively associated with COPD death/hospitalisation in this restricted sample (HR 1.39; 95% CI: 1.02, 1.90), but not in the main analyses (HR 1.15; 95% CI: 1.00, 1.32). Associations with early menopause, parity>3, PCOS/ovarian cysts, ever using oral contraception or hormone replacement therapy, however, became null. We decided to allocate the analyses of never smokers to the sensitivity analyses because of the small number of COPD events, i.e. greatest number in any category was 208 events/159 202 women who had no history of endometriosis while many categories had fewer than 20 events.

Results (p. 11)

“In our analyses of never smokers (n=161 626), however, associations of COPD-related hospitalisation/death with early menopause, parity>3, PCOS/ovarian cysts, ever using oral contraception or hormone replacement therapy became null (online supplementary file Table S5). Age at menarche was also positively associated with COPD-related hospitalisation/death in this restricted sample of never smokers (HR 1.39; 95% CI: 1.02, 1.90), whilst the remaining associations remained unchanged from the main analyses.”

Unfortunately, we are unable to adjust for environmental tobacco smoke as these data are not collected in UK Biobank. We agree with Reviewer 2 that ETS is a key confounding variable and there may be residual bias in our model. We therefore acknowledge this as a limitation in the discussion.

Strengths and limitations of this study (p. 3)

- Data on environmental tobacco smoke were not available and we cannot exclude residual bias from this exposure.

Discussion (p. 15)

“Lastly, we were unable to adjust for environmental tobacco smoke exposure as these data were not collected in UK Biobank. We therefore cannot exclude the possibility of residual bias from environmental tobacco smoke.”

Comment 3: In one of the analysis the study compares lung function and incidence of COPD hospitalization/mortality in post-menopausal women with pre-menopausal women (reference). We know that pre-menopausal women are younger compared to post-menopausal women. I am not sure that just by adjusting for the age is enough? If there is enough sample, I would suggest you to analyze the post- versus pre-menopausal in a very narrow age-band. Alternatively, you could analyse the data for men and women separately and see if women of same age group as men have higher incidence of hospitalization and mortality.

Response 3: We have now analysed the association of menopause status with COPD hospitalisation/death excluding women age <45 years or >60 years at baseline, as the mean age at menopause among UK women is 51 years. Among the 271 271 participants (age range 39-71 years) without COPD at baseline, 143 573 participants were between age 45-60 years (inclusive) at baseline. The association between menopause status and COPD-related hospitalisation/death was positive in this restricted sample (HR 1.57; 95% CI: 1.07, 2.30) while the association was null in the larger cohort of UK Biobank women without COPD at baseline.

We also repeated our linear regression of spirometry measurements in participants between age 45-60 (inclusive) at baseline. Results from these sensitivity analyses were similar to those in the main analyses.

Variable	Multiple-adjusted mean change (95% CI)	
	Full cohort of women age 39-71 years at baseline	Women age 45-60 years at baseline
N	271 271	143 573
FEV ₁	-0.04 (95% CI: -0.05, -0.02)	-0.05 (95% CI: -0.07, -0.03)
FVC	-0.04 (95% CI: -0.06, -0.03)	-0.05 (95% CI: -0.07, -0.03)
FEV ₁ /FVC	0.01 (95% CI: -0.01, 0.02)	0.00 (95% CI: -0.02, 0.02)

Results (p. 9-10)

“After restricting the sample to participants who were age 45-60 years (inclusive) at the time of reporting menopause status to reduce residual bias due to age, we found that menopause status was positively associated with COPD-related hospitalisation/death (HR 1.57; 95% CI: 1.07, 2.30), whilst associations with spirometry measures did not change (results not shown).”

Comment 4: Why do you combine the COPD hospitalization and mortality in one analysis. They are likely to have different grade of COPD and I would separate the analysis if possible.

Response 4: In response to your comment, we have performed Cox proportional hazards regression of COPD mortality. The findings from these analyses were null for all reproductive health indicators, except age at menarche <12 years compared to 12-15 years (HR 1.52; 95% CI: 1.08, 2.13). There were only 179 COPD-related deaths in the study population of 273 441 women, and some exposure categories had fewer than ten events. We therefore had limited power to examine the associations with COPD death despite the large sample size.

Results (p. 11-12)

“We also investigated associations of reproductive health indicators with risk of COPD-related death among women in UK Biobank. The associations from these analyses of 273 441 women (including those with COPD reported at baseline) were null for all reproductive health indicators (online supplementary file Table S9), except age at menarche <12 years compared to 12-15 years (HR 1.52; 95% CI: 1.08, 2.13). However, there were only 179 COPD-related deaths and many categories of our reproductive health indicators experienced fewer than ten events.”

Comment 5: You have reported that early or late menarche are associated with increased risk of COPD-related hospitalization /mortality and same for post-menopause. I was wondering if you can create a score using all the potential factors that you have identified and analyze the result based on it.

Response 5: Do you mean to develop a predictive risk score for COPD hospitalisation/death by multivariable logistic regression using the reproductive health indicators we have found non-zero associations for? Due to the relatively short follow-up period (median 6 years) and the small number of COPD-related hospitalisation/death events, we believe the discrimination performance of a model developed from these data would not be acceptable. Developing such a risk score would be of great value but it is unfortunately outside the scope of the present study due to data limitations.

Comment 6: Your lung function indices findings for different reproductive history might have the same bias due to age. How about analyzing for percent predicted values, this should take account for age and height and also ethnicity.

Response 6: In response to this comment and to Comment 3 from Reviewer 3, we now see that we had not clearly described our conversion of spirometry measures to z-scores standardised by age and height. We had decided on age and height-standardised z-scores calculated as [(observed-predicted)/standard deviation] per the Global Lung Initiative’s 2012 guidance,³ as the percent predicted values have been found to be biased.⁴

Methods (p. 6)

“Spirometry measurements were converted to internally standardised z-scores by age and height based on the Global Lung Initiative 2012 recommendations³ to reduce bias related to age. These z-scores were used for all analyses.”

Methods (p. 8)

“We reported results as mean difference in the spirometry measurements z-scores with 95% CI. These mean differences in z-scores can be converted to raw values by multiplication with the standard deviation in spirometry measurement for the given age and height group.”

Due to the low proportion (5.4%) of non-white ethnicities reported in UK Biobank women, we decided not to standardise by ethnicity, although we did adjust for ethnicity in all multivariable adjusted analyses.

Reviewer: 3

Reviewer Name: Pentti Nieminen

Institution and Country: Medical Informatics and Data Analysis Research Group, University of Oulu, Finland

Please state any competing interests or state 'None declared': None declared

Please leave your comments for the authors below

I enjoyed reading this manuscript. I have reviewed this manuscript with a particular emphasis on the statistical methods and analyses used. It was particularly interesting to see how good the statistical reporting was. I have some observations that should receive the attention of the authors.

Comment 1: The statistical intensity of this manuscript was close to the average of articles published in general medical journals. However, the quality of statistical reporting is very high: 10 in a scale from 0 (poor) to 10 (very high).

Response 1: Thank you for your comment.

Comment 2: Page 6, line 60: Please consider adding a reference to Schoenfeld residuals.

Response 2: We have now added a reference by Schoenfeld⁵ to our Methods (p. 7).

Comment 3: Page 7, line 32: I don't understand why you have analysed z-scores of spirometry measurements. I find z-score difficult to interpret compared to original values that have a clear clinical interpretation. Please help your readers and clarify.

Response 3: In response to your comment and to Comment 6 from Reviewer 2, we have revised our Methods to include an explanation for z-score standardisation of the spirometry measurements by age and height. Standardization of spirometry measurements to z-scores is recommended by the Global Lung Initiative,³ rather than using percent predicted values, since the variability of percent predicted might lead to bias.⁴

Methods (p. 6)

"Spirometry measurements were converted to internally standardised z-scores by age and height based on the Global Lung Initiative 2012 recommendations³ to reduce bias related to age. These z-scores were used for all analyses."

Methods (p. 8)

"We reported results as mean difference in the spirometry measurements z-scores with 95% CI. These mean differences in z-scores can be converted to raw values by multiplication with the standard deviation in spirometry measurement for the given age and height group."

Comment 4: Table 2: Consider adding the method (Cox regression) to the title.

Response 4: We hesitate to expand on our current title "Female reproductive history in relation to chronic obstructive pulmonary disease and lung function in UK Biobank: a prospective population-based cohort study" given its length and risk overwhelming the reader and obscuring the final message.

Comment 5: Table 2, page 26, line14: Clarify what is meant with "ordinal (5)".

Response 5: We appreciate the opportunity to provide further clarification. This has been edited to state "ordered categories" for Tables 2-4 in the Main Document and Tables S3-S11 in the online supplementary file.

Comment 6: Table 3: Consider adding the method (linear regression) to the title.

Response 6: As with our response to Comment 4, we are reluctant to expand on our already lengthy manuscript title. We appreciate your understanding.

Comment 7: Tables 2: Is the total number of analysed subjects 271271 (in the title) or 159571 (footnote b)? Clarify whether you have used complete cases or imputed data in Tables 2 and 3.

Response 7: The number of participants in the title are those analysed for all reproductive health indicators except age at menopause. Footnote b refers to the “Age at natural menopause” reproductive health indicator only. As these data are only available in women who have already experienced menopause, a smaller subset (n=159 571) of the total n=271 271 women were included in analyses of age at menopause.

We have revised the footnotes for Tables 2-3 and online supplementary file Tables S3-11 to state, “Age at menopause was analysed only among women who experienced natural menopause before baseline (n=XXX XXX)”, where XXX XXX differs depending on the analyses. E.g. n=159 571 in the footnote of Table 2 because n=271 271 did not have COPD at baseline, whilst n=127 270 in the footnote of Table S3 because n=217 512 did not have any respiratory illness at baseline or FEV₁/FVC<0.7 at baseline.

VERSION 2 – REVIEW

REVIEWER	Anthony B Miller Dalla Lana School of Public Health, University of Toronto, Canada
REVIEW RETURNED	01-Aug-2019

GENERAL COMMENTS	The reviewer completed the checklist but made no further comments.
--

REVIEWER	Om P Kurmi McMaster University, Canada
REVIEW RETURNED	16-Aug-2019

GENERAL COMMENTS	The authors have addressed majority of the comments/queries and have provided justification where they have not. I do not have any further comments to make.
--

REVIEWER	Pentti Nieminen University of Oulu, Finland
REVIEW RETURNED	28-Jul-2019

GENERAL COMMENTS	I have only one further comment. I now see that I had not clearly formulated my comment about the titles for Tables 2-3. I did not ask to expand on the title of your manuscript, I meant the titles for Tables 2-3. Please consider adding the name of applied regression method to the titles of these tables. Or you can provide this information in the footnotes for Tables 2-3.
---

VERSION 2 – AUTHOR RESPONSE

Reviewer: 3

Reviewer Name: Pentti Nieminen

Institution and Country: University of Oulu, Finland

Please state any competing interests or state 'None declared': None declared

Please leave your comments for the authors below

I have only one further comment. I now see that I had not clearly formulated my comment about the titles for Tables 2-3. I did not ask to expand on the title of your manuscript, I meant the titles for Tables 2-3. Please consider adding the name of applied regression method to the titles of these tables. Or you can provide this information in the footnotes for Tables 2-3.

Response: We have now revised the titles for Tables 2 and 3 in the main document, as well as Tables S3-S11 of the online supplement, to indicate whether “Cox regression analyses” or “Linear regression analyses” were performed. Thank you for your